# Anomalous continuous translations

**Nathan Seiberg**

School of Natural Sciences, Institute for Advanced Study, Princeton, NJ

*Part of the In Memoriam: Ian Affleck Collection*

## Abstract

We discuss a large class of non-relativistic continuum field theories where the Euclidean spatial symmetry of the classical theory is violated in the quantum theory by an Adler-Bell-Jackiw-like anomaly. In particular, the continuous translation symmetry of the classical theory is broken in the quantum theory to a discrete symmetry. Furthermore, that discrete symmetry is extended by an internal symmetry, making it non-Abelian. This presentation streamlines and extends the discussion in [1]. In an Appendix, we present an elementary introduction to 't Hooft and Adler-Bell-Jackiw anomalies using a well-known system.

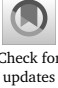

## 1  Introduction

In this note, we will discuss a large class of continuum classical field theories with spatial Euclidean symmetry and time translation symmetry. A common feature of all these models is that the classical Lagrangian contains a subtle term. The proper definition of this term in the quantum theory shows that the classical translation symmetry is explicitly violated in the quantum theory and becomes non-Abelian. We will interpret this violation of the classical symmetry as an Adler-Bell-Jackiw (ABJ)-like anomaly.

Related discussions have appeared in the context of gauge theories with constant background charge [2–4] and ferromagnets [5–16]. Following these authors, the main point of [1] was to trace the peculiar phenomena to a careful definition of the Lagrangian and to interpret the phenomenon as an ABJ-like anomaly. Here, we will take a broader point of view considering different theories related by turning on background field as belonging to the same family. This perspective will allow us to define the Lagrangian using known results about Chern-Simons terms and to obtain a deeper understanding of the violation of continuous translations.

Consider a theory with dynamical fields $\phi^r$. The classical Lagrangian density is $\mathcal{L}_{classical}(\phi^r)$ and the equations of motion are $\frac{\delta}{\delta\phi^s}\mathcal{L}_{classical}(\phi^r) = 0$. Often, the equations of motion are well-defined, but the Lagrangian density $\mathcal{L}_{classical}(\phi^r)$ or even the action $S = \int \mathcal{L}_{classical}$ (or its Euclidean version $S^E = \int \mathcal{L}^E_{classical}$) are not. This is harmless in the classical theory.

In the quantum theory, we can still have ill-defined $S$ (or $S^E$), provided we have a meaningful definition of $\exp\left(\frac{i}{\hbar}S\right)$ (or $\exp\left(-\frac{1}{\hbar}S^E\right)$).[1] This definition, should be such that the equations of motion are as in the classical theory. Hence, the difference between the classical Lagrangian $\mathcal{L}_{classical}$ and the quantum Lagrangian $\mathcal{L}$ do not contribute to the equations of motion. For example, they could be total derivatives.

When comparing the classical theory and the quantum theory, two kinds of phenomena often happen:

- The coefficients of some of the terms in the classical Lagrangian density $\mathcal{L}_{classical}$ should be quantized in units of $\hbar$. This arises because these terms are not well-defined. The quantization of their coefficients ensures that $\exp\left(\frac{i}{\hbar}S\right)$ (or $\exp\left(-\frac{1}{\hbar}S^E\right)$) are well-defined. Familiar examples are the Wess-Zumino term [17] and the Chern-Simons term [18].

---

[1]Since we focus on the distinction between the classical and the quantum theory, we will present $\hbar$ explicitly, rather than setting it to one. See our conventions in Section 1.2.

- There are new terms in the quantum theory that do not affect the classical physics. For example, there can be terms that are locally total derivatives and therefore they do not contribute to the equations of motion. However, globally, they are nontrivial and can contribute to the action. It is common to refer to these terms as $\theta$-terms. (Other $\theta$-terms cannot be expressed as integrals over local terms.)

In the examples below, we will see combinations of these phenomena. First, some coefficients will have to be quantized. And second, they will be accompanied by $\theta$-terms. Furthermore, these $\theta$-terms will not have a natural origin ($\theta = 0$) and they will depend on a choice of origin in space, thus breaking the translation symmetry.

The proper mathematical setting to address these subtle issues uses differential cohomology and pre-quantization. (See e.g., the physics discussion in [1,19–23] and references therein for the mathematics literature, in particular [24–26].) The approach in [1] was less sophisticated and used a choice of local trivialization. Spacetime was covered by patches with transition functions between them and the exponential of the action was defined such that the results are independent of the chosen local trivialization.

Since most physicists are not familiar with this mathematical machinery, we will take here another approach. We will derive some of these conclusions using a fact, which follows from differential cohomology, that most physicists do know. Next, we will review this fact.

## 1.1 A useful mathematical fact

Consider two $U(1)$ gauge fields $A^1$ and $A^2$ of degrees $q^1$ and $q^2$, respectively. Their gauge transformations are

$$
\begin{aligned}
A^1 &\to A^1 + d\lambda^1, \\
A^2 &\to A^2 + d\lambda^2,
\end{aligned}
\tag{1}
$$

where $\lambda^1$ and $\lambda^2$ are locally $q^1 - 1$ and $q^2 - 1$ forms. Globally, they are higher-form $U(1)$ gauge fields.[2] Then, on a $q^1 + q^2 + 1$ dimensional closed (Euclidean) spacetime we would like to consider the Chern-Simons term $CS(A^1, A^2)$.

Ignoring global issues, it can be written as $\frac{1}{2\pi} \oint_{\text{spacetime}} A^1 \wedge dA^2$. When this term is present in the classical Lagrangian, its contributions to the classical equations of motion are well-defined and no special care of the global issues should be taken.

However, in the quantum theory, we need a better definition of $CS(A^1, A^2)$. The proper definition of the Chern-Simons term is well-known and relies on differential cohomology. In some cases, one can use an extension of spacetime to a higher-dimensional bulk and define it as $\frac{1}{2\pi} \int_{\text{bulk}} dA^1 \wedge dA^2$ [18]. We can write it symbolically as

$$
\exp\left(ik\, CS(A^1, A^2)\right) = \exp\left(\frac{ik}{2\pi} \oint_{\text{spacetime}} A^1 \wedge dA^2 + \dots\right).
\tag{2}
$$

This expression is well-defined only for integer $k$. Using the definition with a bulk, the quantization follows from the requirement that the answer is independent of the extension to the bulk [18]. This is an example of the phenomenon mentioned above that in the quantum theory, some classical parameters are quantized in units of $\hbar$.

Let us consider shifting $A^1 \to A^1 + \xi^1$ with $\xi^1$ a $q^1$-form flat gauge field (i.e., $d\xi^1 = 0$). This is $U(1)^{(1)}$ $q^1$-form transformation. If we could integrate by parts in the exponent in $\exp\left(\frac{ik}{2\pi} \oint_{\text{spacetime}} A^1 \wedge dA^2 + \dots\right)$, that expression would have been invariant. However, taking

---

[2]We can take $q^1$ or $q^2$ (or both) to be zero. In that case, the transformations (1) are interpreted as shifts by $2\pi\mathbb{Z}$.

the global issues into account,

$$\exp\left(ikCS\left(A^1,A^2\right)\right) \to \exp\left(\frac{ik}{2\pi}\oint_{\text{spacetime}} \xi^1 \wedge dA^2\right) \exp\left(ikCS\left(A^1,A^2\right)\right). \tag{3}$$

Therefore, $\exp\left(ikCS\left(A^1,A^2\right)\right)$ is invariant only when the holonomies of $\xi^1$ around any $q^1$-cycle $\Sigma^{q^1}$ satisfy

$$\exp\left(ik\oint_{\Sigma^{q^1}} \xi^1\right) = 1. \tag{4}$$

As a result, we have only a $\mathbb{Z}_k^{(1)} \subset U(1)^{(1)}$ $q^1$-form symmetry.

Somewhat less obvious is the fact that we do not have a $U(1)^{(2)}$ $q^2$-form symmetry transformation shifting $A^2 \to A^2 + \xi^2$ with $\xi^2$ a $q^2$-form flat gauge field. This follows from the proper definition of the Chern-Simons term, and in particular the terms denoted by the ellipses in (2). The conclusion is that as in (4), $\exp\left(ikCS\left(A^1,A^2\right)\right)$ is invariant only when the holonomies of $\xi^2$ around any $q^2$-cycle $\Sigma^{q^2}$ satisfy

$$\exp\left(ik\oint_{\Sigma^{q^2}} \xi^2\right) = 1. \tag{5}$$

As a result, we have only a $\mathbb{Z}_k^{(2)} \subset U(1)^{(2)}$ $q^2$-form symmetry.

The fact that the naive $U(1)^{(1)} \times U(1)^{(2)}$ higher-form symmetry of $\exp\left(ikCS\left(A^1,A^2\right)\right)$ is explicitly broken to a $\mathbb{Z}_k^{(1)} \times \mathbb{Z}_k^{(2)}$ higher-form symmetry will be crucial below.

The simplest example of such a Chern-Simons term arises for $q^1 = q^2 = 0$, where $A^r$ are circle-valued scalars $A^r \sim A^r + 2\pi$ (see footnote 2). They parameterize a two-torus. When $A^r$ are dynamical fields, which we denote by lower-case characters $\phi^r$, the quantum theory based on $\exp\left(ikCS(\phi^1,\phi^2)\right)$ is known as the non-commutative torus or the fuzzy torus. Physically, it describes a particle on a torus with constant magnetic field with flux $k$. Its global symmetry is known to be $\mathbb{Z}_k^{(1)} \times \mathbb{Z}_k^{(2)}$. We will return to this case in appendix A.

## 1.2 Our conventions

Throughout this note, we will study theories in $d$ spatial dimensions. Our notation is that Lorentzian signature time is denoted by $t$ and Euclidean signature time is denoted by $\tau$. The spatial indices are denoted by $i, j, \ldots$ and the spacetime indices are denoted by $\mu = t, i, j, \ldots$ or $\mu = \tau, i, j, \ldots$.

Since we are interested in studying translations, we will take space to be a flat $d$-dimensional torus $\mathbb{T}^d$ parameterized by

$$x^i \sim x^i + \ell^i, \tag{6}$$

with volume

$$V = \prod_i \ell^i. \tag{7}$$

Our conventions are such that we contract the spatial indices with $\delta_{ij}$. Therefore, we do not distinguish between upper and lower spatial indices.

We will use lower-case characters to denote dynamical fields, e.g., a dynamical $U(1)$ gauge field $a$ or a scalar field $\phi$, and upper-case characters to denote classical background fields, e.g., $A$.

We will use form notation both in the target space and in spacetime, but will also use index notation. For example, a dynamical $U(1)$ gauge field is denoted as

$$a = \sum_\mu a_\mu dx^\mu, \tag{8}$$

and the field strength is

$$f = da = \frac{1}{2} \sum_{\mu\nu} f_{\mu\nu} dx^\mu \wedge dx^\nu, \tag{9}$$

$$f_{\mu\nu} = \partial_\mu a_\nu - \partial_\nu a_\mu.$$

Our gauge fields have standard normalization, e.g., for $U(1)$ gauge fields,

$$\frac{1}{2\pi} \oint dA \in \mathbb{Z}. \tag{10}$$

Our currents are forms with the conservation equation being $dj = 0$. We normalize them such that their charges are quantized

$$\mathcal{Q} = \oint j \in \mathbb{Z}. \tag{11}$$

With these conventions, the coupling of the current $j$ to a gauge field $A$ is through the action term $\hbar \oint_{\text{spacetime}} A \wedge j$, such that the contribution to the functional integral is

$$\exp\left(i \oint_{\text{spacetime}} A \wedge j\right). \tag{12}$$

### 1.3 Outline

In Section 2, we will present the models we study and the relations between them. We will demonstrate the general abstract presentation using two classes of examples, $U(1)$ gauge theories and certain non-linear sigma models. In Section 3, we will deform these models by coupling them to a particular background gauge field.

Section 4 will derive the ABJ-like anomaly in continuous translations. And in Section 5, we will use this understanding to perform a non-trivial check of some dualities.

Section 6 will summarize our results and will review some related topics from [1].

Appendix A will be devoted to a pedagogical example of the notion of 't Hooft anomalies and ABJ anomalies in the context of a particle in a magnetic field. That appendix does not have a single new result. Instead, it presents this well-known physics from the perspective of symmetries, their anomalies, and their gauging. Also, the symmetry discussed in that appendix is closely related to the translation symmetry in the body of this paper.

## 2 The theories

The model we study are characterized by having a two-form current $j$ with quantized periods

$$\mathcal{Q}_{\Sigma^2} = \oint_{\Sigma^2} j \in \mathbb{Z}. \tag{13}$$

Furthermore, we demand that the condition (13) is satisfied off-shell, i.e., without using the equations of motion.

An alternative way to characterize these theories is to say that $j$ is the current of a $d-2$-form $U(1)$ symmetry with charge $\mathcal{Q}_{\Sigma^2}$ [27]. For $d \geq 2$, the condition (13) shows that the symmetry operator $\mathcal{Q}_{\Sigma^2}$ is topological and hence, the current $j$ is conserved, i.e., $dj = 0$. (We demand that this equation is valid off-shell.) For $d = 1$, this is a $-1$-form symmetry [22,27,28]. It is meaningless to say that the charge is topological and that $j$ is conserved. Yet, we can still consider the condition (13) and demand that it is satisfied off-shell.

Since we imposed (13) (and therefore for $d \geq 2$ also the conservation $dj = 0$) off-shell, we have locally

$$j = d\varphi. \tag{14}$$

Importantly, $\varphi$ is not-well defined. Only its derivative (14) is well defined.

A common feature of all these models is that the configuration space includes disconnected regions labeled by $\mathcal{Q}_{\Sigma^2} = \oint_{\Sigma^2} j$. Consequently, in the quantum theory, we can write $\theta$-terms $\theta j$. For $d = 1$, this is a standard $\theta$-term. For $d > 1$, such a term is not Euclidean invariant. But if our space is a torus with coordinates $x^i \sim x^i + \ell^i$, we can include in the Euclidean functional integral a factor depending on $d$ $\theta$-parameters

$$\exp\left( i \sum_i \theta^i \oint d\tau dx^i \, j_{\tau i} \right) = \exp\left( i \sum_i \theta^i \frac{\ell^i}{V} \oint d\tau d^d x \, j_{\tau i} \right), \qquad V = \ell^1 \ell^2 \cdots \ell^d. \tag{15}$$

Since this characterization of the theories might be abstract, let us consider some examples.

## 2.1 Example 1: $U(1)$ gauge theory

Any field theory with a $U(1)$ gauge field $a$ satisfies these conditions. The two-form current $j$ and $\varphi$ are recognized as

$$\begin{aligned} j &= \frac{1}{2\pi} f = \frac{1}{2\pi} da, \\ \varphi &= \frac{1}{2\pi} a. \end{aligned} \tag{16}$$

In this case, the $d-2$-form symmetry is known as the magnetic symmetry.

## 2.2 Example 2: Some non-linear sigma-models

Another class of examples arises when the theory includes fields taking values in a target space $\mathcal{M}$ with nonzero second de Rham cohomology with quantized periods. Then, for $\omega \in H^2(\mathcal{M}, \mathbb{R})$ with $\int \omega \in \mathbb{Z}$ we take $j$ as its pullback to spacetime and locally $j = d\varphi$. In fact, these examples can be presented as the previous case (16) as follows. We let the target space be a circle bundle over $\mathcal{M}$ with $c_1 = \omega$ and then add a gauge field $a$ to remove this added circle.

Let us demonstrate these models using the well known $SO(3)$ sigma-model. We parameterize the $S^2$ target space by a complex (stereographic) coordinate $z$. Then,

$$\begin{aligned} j &= \frac{i}{2\pi} \frac{dz \wedge d\bar{z}}{(1+|z|^2)^2}, \\ \varphi &= \frac{i}{2\pi} \frac{z d\bar{z} - \bar{z} dz}{1+|z|^2}. \end{aligned} \tag{17}$$

In this case, it is common to present the theory as a $U(1)$ gauge theory coupled to an $S^3$ target space. (This is usually referred to as the $CP^1$ presentation of the theory.) Then, the expressions (17) can be replaced by (16).

## 3  Coupling to a background gauge field $A$

Given that our theories have a $d-2$-form $U(1)$ symmetry it is natural to couple them to a background gauge field for that symmetry. This is a $d-1$-form $U(1)$ gauge field $A$. Its coupling is through the factor of

$$\exp\left(i\oint_{\text{spacetime}} A\wedge j\right),\tag{18}$$

in the functional integral. (Recall our conventions about the normalizations in Section 1.2.)

We will be particularly interested in topologically nontrivial $A$. Then, the coupling (18) needs to be defined more carefully. Since we wrote $j = d\varphi$, it is clear how we should do it. We couple $A$ to the theory by inserting in the functional integral (see (2))

$$\begin{aligned}\exp\left(iCS(A, 2\pi\varphi)\right) &= \exp\left(i\oint_{\text{spacetime}} A\wedge d\varphi + \ldots\right)\\ &= \exp\left(i\oint_{\text{spacetime}} \varphi\wedge dA + \ldots\right).\end{aligned}\tag{19}$$

For $d = 1$, our symmetry is a $-1$-form symmetry and its "background gauge field" is the $\theta$-parameter. Then, the coupling (19) is

$$\begin{aligned}\exp\left(iCS(\theta, 2\pi\varphi)\right) &= \exp\left(i\oint_{\text{spacetime}} \theta d\varphi + \ldots\right)\\ &= \exp\left(i\oint_{\text{spacetime}} \varphi\wedge d\theta + \ldots\right).\end{aligned}\tag{20}$$

This means that we turn on arbitrary spacetime dependent $\theta$-parameter. (Compare with (15), which has spacetime independent $\theta$.)

For $d = 2$, our global symmetry is a 0-form symmetry and its background gauge field is a standard $U(1)$ background field. This means that we place our system in a standard background gauge field.

Next, we consider a specific background gauge field $A$ with constant "magnetic field"

$$dA = \frac{2\pi k}{V}dx^1\wedge dx^2\wedge\cdots dx^d.\tag{21}$$

The parameter $k$ is quantized to guarantee

$$\oint_{\text{space}} dA = 2\pi k \in 2\pi\mathbb{Z}.\tag{22}$$

(Note that this is a magnetic field for the background field $A$ not for the dynamical field $a$.)

This background $A$ is topologically nontrivial and therefore it is subtle. The expression (19) becomes

$$\exp\left(iCS(A, 2\pi\varphi)\right) = \exp\left(\frac{2\pi i k}{V}\oint_{\text{spacetime}} d\tau d^d x\,(\varphi_\tau + \ldots)\right).\tag{23}$$

The first term in the right-hand side is a contribution to the Euclidean classical Lagrangian density $\frac{2\pi i k}{V}\varphi_\tau$. And the ellipses represent "correction terms" needed to make it well defined.

Let us turn to the two examples in Section 2.

## 3.1 Example 1: $U(1)$ gauge theory

Here, the term (23) is

$$\exp\Big(iCS(A,a)\Big) = \exp\left(\frac{ik}{V} \oint_{\text{spacetime}} d\tau d^d x \,(a_\tau + \dots)\right). \tag{24}$$

Classically, it affects only the equation of motion of $a_\tau$, i.e., Gauss's law. This constrains the charge density to be $\rho = \frac{\hbar k}{V}$. Quantum mechanically, we see an example of the first subtlety in quantization in Section 1, where $\rho$ is quantized in units of $\hbar$. Physically, this quantization arises because the total charge of the system $\rho V = \hbar k$ should be quantized.

This model with fixed charge density was discussed in [2–4].

For $d = 1$, we have the $\theta$-term (19)

$$\begin{aligned}
\exp\Big(iCS(\theta,a)\Big) &= \exp\left(\frac{i}{2\pi} \oint_{\text{spacetime}} \theta \, da + \dots\right) \\
&= \exp\left(\frac{i}{2\pi} \oint_{\text{spacetime}} a \wedge d\theta + \dots\right).
\end{aligned} \tag{25}$$

The background (21) is $d\theta = \frac{2\pi k}{\ell^1} dx^1$. Locally, we can write

$$\theta = 2\pi k \frac{x^1 - x_0^1}{\ell^1}, \tag{26}$$

with a constant $x_0^1$ whose significance will be discussed below. Then, we find the $d = 1$ version of (24).

We conclude that the $\theta$-term (15), (20), (25) with $\theta$ linear in $x^1$ leads to the model with constant charge density.

## 3.2 Example 2: Some non-linear sigma-models

In this case, the term (23) is first order in time derivative. Such a term is known as a Berry-term or a Wess-Zumino term.

Of particular interest is the case where the target space $\mathcal{M}$ is a symplectic manifold and the two-form $\omega$ is the symplectic structure. In this case $\varphi$ is the pull-back of the Liouville form. By definition, the symplectic structure is non-degenerate. Consequently, at low energies, the term (23) is always more important than any higher time derivative term. Therefore, we can neglect the higher time derivative terms and the low-energy theory includes only a single time derivative term (and of course, higher spatial derivative terms and potential terms).

For example, for the $SO(3)$ sigma-model, $j$ and $\varphi$ are given by (17). The term (23) leads to

$$\exp\Big(iCS(A,2\pi\varphi)\Big) = \exp\left(-\frac{k}{V} \oint_{\text{spacetime}} d\tau d^d x \left(\frac{z\partial_\tau \bar{z} - \bar{z}\partial_\tau z}{1 + |z|^2} + \dots\right)\right). \tag{27}$$

And since the $S^2$ target space is symplectic, we can neglect all terms with more than one time derivative. The resulting Lagrangian is the known continuum Lagrangian for a ferromagnet.

An anti-ferromagnet is described by the $SO(3)$ sigma model with $k = 0$. We now recognize the continuum description of a ferromagnet as the same theory in a background $A$ with constant magnetic field $dA$.

For $d = 1$, the $SO(3)$ sigma-model has a $\theta$-term. As in the $U(1)$ gauge theory discussion, for $\theta = 2\pi k \frac{x^1 - x_0^1}{\ell^1}$, we find the term (27), which describes a ferromagnet.

# 4 Anomalous translations

The constant magnetic field (21) does not determine $A$ uniquely. First, we have freedom in gauge transformations. If there are no 't Hooft anomalies associated with the symmetry, this gauge dependence does not affect the answer. However, the holonomies of $A$ can include additional information. We can parameterize it as

$$H^i(A) = \exp\left( i \oint_{\Sigma^i} A + \dots \right) = \exp\left( 2\pi i k (-1)^{i+1} \frac{x^i - x_0^i}{\ell^i} \right), \tag{28}$$

where $\Sigma^i$ is the $d-1$-dimensional torus with coordinates $x^j \sim x^j + \ell^j$ with $j \neq i$ and $x_0^i$ are constants.

For $d = 1$, the "holonomy" (28) is $H^1(A) = \exp\left(i\theta(x^1)\right) = \exp\left(2\pi i k \frac{x^1 - x_0^1}{\ell^1}\right)$. For $d = 2$, the ellipses in (28) represent contributions from the transition functions leading to $H^1(A) = \exp\left(2\pi i k \frac{x^1 - x_0^1}{\ell^1}\right)$ and $H^2(A) = \exp\left(-2\pi i k \frac{x^2 - x_0^2}{\ell^2}\right)$.

As we will discuss, the physical answers depend on the $d$ constants $x_0^i$. These constants can be changed by shifting $A$ by a constant. Alternatively, they can be interpreted as a choice of a point in space. The dependence on this choice signals the breaking of the continuous translation symmetry.

In [1], a careful analysis of

$$\exp\left( \frac{2\pi i k}{V} \oint d\tau d^d x \, (\varphi_\tau + \dots) \right), \tag{29}$$

determined its $x_0^i$ dependence. That analysis was intrinsic to this term.

Our discussion here circles around the background field $A$ and then (29) is interpreted as a Chern-Simons term

$$\begin{aligned}
\exp\left( iCS(A, 2\pi\varphi) \right) &= \exp\left( i \oint_{\text{spacetime}} A \wedge d\varphi + \dots \right) \\
&= \exp\left( \frac{2\pi i k}{V} \oint d\tau d^d x \, \varphi_\tau + \dots \right).
\end{aligned} \tag{30}$$

The dependence on the constant mode of $A$ is not manifest in the last expression in (30). However, as we discussed in section 1.1, that dependence is present when this term is properly defined as a Chern-Simons term.

More quantitatively, a translation transformation $x^i \to x^i + \epsilon^i$ maps the holonomies (28) as

$$H^i(A) \to \exp\left( 2\pi i k (-1)^{i+1} \frac{\epsilon^i}{\ell^i} \right) H^i(A). \tag{31}$$

This is the same as shifting $A$ by a constant gauge field

$$A \to A + 2\pi k (-1)^{i+1} \frac{\epsilon^i}{V} dx^1 \wedge \cdots dx^{i-1} \wedge dx^{i+1} \cdots \wedge dx^d, \tag{32}$$

without changing the transition functions.

As in (3), this has the effect of mapping

$$\begin{aligned}
\exp\left( iCS(A, 2\pi\varphi) \right) &\to \exp\left( 2\pi i \frac{k\epsilon^i}{V} \oint d\tau d^d x \, j_{\tau i} \right) \exp\left( iCS(A, 2\pi\varphi) \right) \\
&= \exp\left( 2\pi i \frac{k\epsilon^i}{\ell^i} \oint d\tau dx^i \, j_{\tau i} \right) \exp\left( iCS(A, 2\pi\varphi) \right).
\end{aligned} \tag{33}$$

Comparing with (15)

$$\exp\left(i \sum_i \theta^i \oint d\tau dx^i \, j_{\tau i}\right) = \exp\left(i \sum_i \theta^i \frac{\ell^i}{V} \oint d\tau d^d x \, j_{\tau i}\right), \tag{34}$$

we see that $x^i \to x^i + \epsilon^i$ acts the same as

$$\theta^i \to \theta^i + 2\pi \frac{k\epsilon^i}{\ell^i}. \tag{35}$$

The conclusion of this discussion is that the translation symmetry of the classical theory is violated in the quantum theory. We interpret this explicit breaking as due to an ABJ-like-anomaly. We would like to make several comments about this breaking, highlighting the analogy with the ABJ-anomaly of the chiral symmetry.

- As in the chiral anomaly, the anomalous continuous symmetry shifts a $\theta$-parameter (35). Related to that, the symmetry is violated by instantons – configurations with nonzero $\oint d\tau dx^i j_{\tau i} \in \mathbb{Z}$.

- For $\epsilon^i = \frac{\ell^i}{k}$, the shift of the $\theta$-parameter does not affect the physics. Therefore, each $U(1)$ translation symmetry is broken to $\mathbb{Z}_k$. We will denote the discrete symmetry transformations implementing $x^i \to x^i + \frac{\ell^i}{k}$ by $T^i$. They satisfy

$$(T^i)^k = 1. \tag{36}$$

- Like the chiral anomaly, this symmetry breaking is explicit breaking rather than spontaneous breaking. Spontaneous breaking leads to Goldstone bosons. In this case, where continuous translation is broken to a discrete subgroup, that would be phonons. However, our breaking is explicit and there are no phonons.

- One way to see that this is indeed explicit breaking is to note that the definition of the theory depends on a choice of a point in space, the parameters $x_0^i$. This choice breaks the symmetry. Related to that, this choice can be changed by shifting the parameters $\theta^i$.

- The situation is thus analogous to that of the $\eta'$ particle. It gets its mass from instantons associated with the chiral anomaly in QCD.

- Unlike the ABJ-anomaly, this anomaly is not in an internal symmetry, but in a spatial symmetry. To underscore this fact, we can consider the theory on a $d$-dimensional sphere (rather than $\mathbb{T}^d$). We can still use a background $A$ with constant magnetic field $dA$. And then, the classical translation symmetry is replaced by a rotation symmetry. However, in this case, there are no $\theta$-terms like (34). Related to that, the discussion of the shift of $A$ cannot be repeated. Equivalently, the analysis in [1] also does not lead to a subtlety in the patches. We conclude that in this case, the rotation symmetry is not violated.

Consider the system in two dimensions. Then, $A$ is an ordinary $U(1)$ gauge field and the constant $dA$ is a constant magnetic field. It is well known that a charged particle on a torus with constant magnetic field with flux $k$ does not have continuous translation symmetry. (See the review in Appendix A.) Its translation symmetry is $\mathbb{Z}_k \times \mathbb{Z}_k$. Furthermore, that translation symmetry is realized projectively. Our system does not have a single particle. Instead, it has excitations with different charges $\mathcal{Q} = \oint_{\mathbb{T}^2} j$ under the background gauge field $A$. As a result, the $\mathbb{Z}_k \times \mathbb{Z}_k$ symmetry is not extended by a c-number, but by an operator. The symmetry algebra is

$$(T^1)^k = (T^2)^k = 1,$$
$$T^1 T^2 = e^{2\pi i \frac{\mathcal{Q}}{k}} T^2 T^1. \tag{37}$$

Since this extension is not central, this is not an 't Hooft anomaly.

The result (37) about $d = 2$ is easily extended to higher dimensions. Focusing on two dimensions at a time, we have

$$
(T^i)^k = 1 ,
$$
$$
T^i T^j = e^{2\pi i \frac{\mathcal{Q}_{ij}}{k}} T^j T^i ,
$$
$$
\mathcal{Q}_{ij} = \oint_{\Sigma_{ij}} j , \tag{38}
$$

where $\Sigma_{ij}$ is the two-torus with $x^i \sim x^i + \ell^i$ and $x^j \sim x^j + \ell^j$.

We conclude that the continuous translation symmetry is not only broken to a discrete subgroup, but it is also extended and becomes non-Abelian.

In the specific context of the two examples of $U(1)$ gauge theory with constant charge density (Section 3.1) and the $SO(3)$ ferromagnet (Section 3.2), this breaking of continuous translation has already been discussed in [2–4] and [5–16] respectively. Our presentation unifies these examples into a much larger set of cases. It also resolves some puzzles and identifies the source of the effect as an ABJ-like anomaly associated with the definition of the integrand in the function integral.

# 5  Checking dualities

Our discussion leads to interesting tests and further insights into dualities. In this context, a dual pair of theories are different classical Lagrangians leading to the same IR quantum physics. Such dualities are particularly interesting when the fields in the two sides of the duality are not related to each other via a local change of variables.

A characteristic example is the famous particle-vortex duality in $2 + 1$ dimensions [29, 30]. It relates the complex $|\phi|^4$ theory to a gauged version of it. Following the discussion in [31], we express this duality as

$$
|D_A \phi|^2 - |\phi|^4 \qquad \longleftrightarrow \qquad |D_a \chi|^2 - |\chi|^4 + \frac{\hbar}{2\pi} a dA. \tag{39}
$$

Here $\phi$ and $\chi$ are complex scalar fields, $a$ is a dynamical $U(1)$ gauge field and $A$ is a background $U(1)$ gauge field. $D_A$ and $D_a$ are covariant derivatives acting on charge $+1$ fields. (We slightly abuse the notation here, where all the terms except the last one are scalars and the last term is a 3-form.)

Both sides of the duality have a global $U(1)$ symmetry coupled to the background gauge field $A$. In the left-hand side, the corresponding current is $j = \frac{1}{\hbar} {}^* (i \bar{\phi} d\phi - id\bar{\phi} \phi + 2A|\phi|^2)$, where we followed the conventions of Section 1.2, i.e., $j$ is a two-form, normalized to have quantized charges. $j$ is conserved only by using the equations of motion, i.e., it is conserved only on-shell. In the right-hand side of the duality, the current is $j = \frac{1}{2\pi} da$ and it is conserved off-shell – it is topological.

Most of the tests of this duality have involved topologically trivial background $A$. However, the duality should be valid for all background $A$. In particular, it should be valid for $A$ of constant magnetic field as in (21). It is nice to see that this is indeed the case.

In the left-hand side of (39), we have charged particles in a constant magnetic field. It is known that the translation symmetry is as in (37). (Again, this is not a central extension of $\mathbb{Z}_k \times \mathbb{Z}_k$.) In the right-hand side of (39), this background $A$ leads to constant charge density. As we have been discussion, the classical theory is translation invariant, but in the quantum theory the symmetry is discrete as in (37).

Similar reasoning applies to related particle/vortex dualities including the fermion/fermion duality of [4, 31–34]. See in particular the discussion in [4]. Our general presentation that depends on the coupling to the current $j$ makes this test of the duality straightforward. In doing it, it is essential to use properly normalized Chern-Simons terms, as in [31].

We see here that for nontrivial $A$ the symmetries in the two sides of the duality do not match in the classical theory, but they do match in the quantum theory. This is quite common. In particular, in many of the electric/magnetic dualities with $N = 1$ supersymmetry in $3 + 1$ dimensions [35,36] the classical symmetries do not match between the two sides of the duality. However, in the quantum theory, some of these symmetries suffer from an ABJ anomaly. Then, only the symmetries without that anomaly match.

## 6   Conclusions

In this discussion we adopted a modern view about quantum field theory, where one turns on all possible background fields and studies the physics as a function of them. We focused on systems with a $U(1)$ $d - 2$-form global symmetry whose conserved current $j$ is conserved off-shell. Then, locally $j = d\varphi$. Coupling it to a background gauge field $A$ is locally of the form

$$\varphi \wedge dA. \tag{40}$$

Globally, we defined this term as a Chern-Simons term

$$
\begin{aligned}
\exp\left(iCS(A, 2\pi\varphi)\right) &= \exp\left(i \oint_{\text{spacetime}} A \wedge d\varphi + \dots\right) \\
&= \exp\left(i \oint_{\text{spacetime}} \varphi \wedge dA + \dots\right).
\end{aligned}
\tag{41}
$$

Then, we specialized to a particular background $A$ with constant magnetic field

$$dA = \frac{2\pi k}{V} dx^1 \wedge dx^2 \wedge \cdots dx^d. \tag{42}$$

This leads to the term

$$\frac{2\pi \hbar k}{V} \varphi_t + \dots \tag{43}$$

in Lagrangian density.

Using this procedure we learned that different theories are unified into the same class. One example is $U(1)$ gauge theories with or without background charge density. Another example is the continuum descriptions of an antiferromagnet and for a ferromagnet. (These different examples are also unified when the latter is presented using a larger target space.)

The coupling to the background field (40) allowed us to view the term (43) as a Chern-Simons term (41) and to use the known precise definition of that term. In fact, all we needed was one fact about it, reviewed in Section 1.1, that globally, the term (40) depends on the constant mode of $A$. Through this dependence, our coupling (43) is not translation invariant.

This subtlety in (43) does not affect the classical theory. It appears through the definition of the integrand in the functional integral. We interpreted this phenomenon as an ABJ-like anomaly in translations.

In the remaining of this section, we will review some aspects of this anomaly that have been discussed in [1].

## 6.1 The anomalous momentum current

Since the classical theory has continuous translation symmetry, we can follow Noether procedure to find a conserved momentum current. It is of the form

$$
\Theta_{tj} = \Theta_{tj}^{(0)} + \frac{2\pi\hbar k}{V}\varphi_j \,,
$$

$$
\Theta_{ij} = \Theta_{ij}^{(0)} + \delta_{ij}\frac{2\pi\hbar k}{V}\varphi_t \,,
\tag{44}
$$

$$
\partial_t\Theta_{tj}^{(0)} - \sum_i \partial^i\Theta_{ij}^{(0)} = \frac{2\pi\hbar k}{V}j_{jt} \,.
$$

The operators $\Theta_{tj}^{(0)}$ and $\Theta_{ij}^{(0)}$ are locally well-defined. But they do not lead to a conserved current. The operators $\Theta_{tj}$ and $\Theta_{ij}$ are conserved, but are not well-defined. This is a familiar situation with the ABJ anomaly.

Using this current, we can construct the discrete translation operator

$$
T^j = \exp\left(\frac{i\ell^j}{\hbar k}\int d^d x \left(\Theta_{tj}^{(0)} + \frac{2\pi\hbar k}{V}\varphi_j + \dots\right)\right).
\tag{45}
$$

And again, this expression has to be defined carefully [1].

## 6.2 Noninvertible continuous translations

In some cases, the system also has a topological $U(1)^1 \times U(1)^2$ $d-1$-form global symmetry with a conserved currents $j^1$ and $j^2$, such that our $d-2$-form global symmetry is derived from it and its current is

$$
j = j^1 \wedge j^2 \,.
\tag{46}
$$

In this case, we can restrict attention to the subspace of the Hilbert space with $\oint j^1 = \oint j^2 = 0$. In this subspace, $\mathcal{Q} = \oint j = 0$ and there are no instantons. Therefore, the continuous translation symmetry is unbroken. However, since the continuous symmetry is valid only in a subspace, it is a noninvertible symmetry [1]. A similar phenomenon has been seen in some cases with an ABJ anomaly [37–40].

## 6.3 The large volume limit

A peculiar aspect of the term (43) is that its coefficient depends explicitly on the total volume $V$. What happens as we take $V \to \infty$? In order to have a nontrivial effect, we should also scale $k \to \infty$ such that $\frac{k}{V}$ is finite. Then, the resulting translation symmetry depends on how we take $V \to \infty$ [1].

For $d = 1$, there is only one way to do it. Then, since for finite $\ell = V$, we have only translations by $\frac{\ell}{k}$, in the limit, we still have only discrete translations, i.e., the translation symmetry is $\mathbb{Z}$.

For $d = 2$, we can take $k \sim \ell^1 \to \infty$ with finite $\ell^2$. Then, the translation in direction 1 becomes $\mathbb{Z}$ and in direction 2, it becomes $U(1)$. We can also take both $\ell^1$ and $\ell^2$ to infinity with $k \sim \ell^1\ell^2$. Then, the symmetry becomes $\mathbb{R}$ in each direction. Either way, the symmetry is still extended by $\mathcal{Q}$.

For higher $d$, there are more options. Depending on the limit, translation in each direction becomes $U(1)$, $\mathbb{R}$ or $\mathbb{Z}$.

## 6.4 Relation to discrete translations of an underlying lattice model

These continuum models can arise as the low-energy approximations of lattice models. For simplicity, consider a cubic lattice with $L^i$ sites in direction $i$. In terms of the lattice spacing $a$, we have $\ell^i = a L^i$. The lattice translation symmetry is generated by $T_{UV}^i$ satisfying $(T_{UV}^i)^{L^i} = 1$. In many cases, the lattice model does not have the $U(1)$ $d-2$-form symmetry of the continuum theory with charge $\mathcal{Q}$ and therefore, the lattice generators $T_{UV}^i$ commute.

What is the relation between this discrete Abelian translation symmetry and the discrete non-Abelian translation symmetry of the continuum theory?

The analysis in [1] led to the conclusion that the lattice generators $T_{UV}^i$ flow as

$$T_{UV}^i \to (T^i)^{\frac{k}{L^i}}, \tag{47}$$

which relies on the fact that $\frac{k}{L^i}$ is an integer. The map (47) means that the lattice translations map to a $\mathbb{Z}_{L^1} \times \mathbb{Z}_{L^2} \cdots \mathbb{Z}_{L^d}$ subgroup of the translation symmetry of the continuum theory. In particular, this subgroup is Abelian. In the context of the $d = 1$ ferromagnet, this result has already appeared in [6].

In most standard cases, the continuum theory has a much larger symmetry than the lattice model. It has continuous translation symmetry and the discrete symmetry of the underlying lattice model is a subgroup of it. The same is true in our case, except that the symmetry of the continuum theory is also discrete (and non-Abelian).

## 6.5 Relation to the Lieb-Schultz-Mattis theorem

The celebrated Lieb-Schultz-Mattis (LSM) theorem [41, 42] can be phrased as a consequence of an 't Hooft anomaly between lattice translations and an internal symmetry of a lattice model [43–51]. Given that continuum translations do not have an 't Hooft anomaly, it is natural to ask how the LSM anomaly is matched in the continuum.

In some cases, a new internal symmetry emanates from lattice translation [47, 51] and the LSM lattice anomaly becomes an ordinary 't Hooft anomaly between the lattice internal symmetry and this emanant internal symmetry. See also [52, 53].

However, this cannot be the whole story. While this description is correct for an antiferromagnet, it cannot be correct for a ferromagnet [1]. One reason for that is that the continuum Lagrangian for a ferromagnet lacks this emanant global symmetry. Another reason is that this continuum Lagrangian does not depend on whether the underlying lattice model does or does not have the LSM anomaly.

The resolution of the puzzle is that while continuous translations cannot have an 't Hooft anomaly, discrete translations can have it. And the map (47) between lattice translations and the continuum translations is compatible with the LSM anomaly [1].

## Acknowledgments

We are grateful to Ofer Aharony, Tom Banks, Dan Freed, Helmut Hofer, Steve Kivelson, Juan Maldacena, Max Metlitski, Gregory Moore, Abhinav Prem, Shinsei Ryu, Sahand Seifnashri, Douglas Stanford, Ryan Thorngren, Senthil Todadri, Shu-Heng Shao, Steve Shenker, Wilbur Shirley, Dam Son, Nikita Sopenko, Yuji Tachikawa, Ashvin Vishwanath, and Edward Witten for useful discussions.

**Funding information** This work was supported in part by DOE grant DE-SC0009988 and by the Simons Collaboration on Ultra-Quantum Matter, which is a grant from the Simons Foundation (651444, NS).

# A  An anomalous view of a charged particle in a constant magnetic field

The purpose of this Appendix is to present a pedagogical example of 't Hooft and Adler-Bell-Jackiw (ABJ) anomalies. We will discuss the well-known problem of a charged particle in a constant magnetic field.

 We will not present any new result. However, this perspective of the system will allow us to draw some general lessons about anomalies and to address some common misconceptions. Also, the symmetry issues that we will discuss are used in the more complicated cases in the body of this note.

 As in the rest of this note, in order to highlight the distinction between classical and quantum effects, we will not set $\hbar$ to one.

## A.1  A charged particle in $\mathbb{R}^2$ with a constant magnetic field

### A.1.1  A classical particle

Consider a charged particle in $\mathbb{R}^2$, parameterized by $(x^1, x^2)$, interacting with a constant magnetic field $B$. Suppressing the electric charge, the classical Lagrangian is

$$\mathcal{L}_{classical} = \frac{m}{2}\left((\partial_t x^1)^2 + (\partial_t x^2)^2\right) + B x^1 \partial_t x^2. \tag{A.1}$$

Clearly, it is not translation invariant in $x^1$. However, translating $x^1 \to x^1 + \epsilon^1$, it is shifted by a total derivative $\mathcal{L}_{classical} \to \mathcal{L}_{classical} + B\epsilon^1 \partial_t x^2$ and therefore, the classical equations of motion

$$\begin{aligned} m\partial_t^2 x^1 &= B\partial_t x^2, \\ m\partial_t^2 x^2 &= -B\partial_t x^1, \end{aligned} \tag{A.2}$$

are translation invariant.

 We conclude that the global symmetry[3] of the system includes

$$G = \mathbb{R} \times \mathbb{R}. \tag{A.3}$$

Using the equations of motion (A.2), the conserved charges are

$$\begin{aligned} p_1 &= m\partial_t x^1 - B x^2, \\ p_2 &= m\partial_t x^2 + B x^1. \end{aligned} \tag{A.4}$$

Importantly, $p_1$ and $p_2$ are not the momenta conjugate to $x^1$ and $x^2$.

### A.1.2  A quantum particle

Since the Lagrangian (A.2) is well-defined, the quantization is straightforward. As is well-known, the conserved charges $p_1$ and $p_2$ of (A.4) do not commute

$$[p_1, p_2] = -i\hbar B. \tag{A.5}$$

Hence, the global symmetry $G = \mathbb{R} \times \mathbb{R}$ of (A.3) is realized projectively. In particular,

$$\begin{aligned} U^1(\epsilon^1)U^2(\epsilon^2) &= \exp\left(\frac{iB\epsilon^1\epsilon^2}{\hbar}\right)U^2(\epsilon^1)U^1(\epsilon^2), \\ U^i(\epsilon^i) &= \exp\left(\frac{i\epsilon^i p_i}{\hbar}\right). \end{aligned} \tag{A.6}$$

---

[3]Note that unlike in the body of this note where translations act on the coordinates in a field theory, here, $x^1$ and $x^2$ parameterize the target space and hence $G$ is an internal symmetry.

In modern terms, we interpret this projective realization of the global symmetry as an 't Hooft anomaly in $G = \mathbb{R} \times \mathbb{R}$.

In order to relate to other presentations of 't Hooft anomalies, we couple the $G = \mathbb{R} \times \mathbb{R}$ global symmetry to two background $\mathbb{R}$ gauge fields $A_t^i$. We let $\epsilon^i$ in $x^i \to x^i + \epsilon^i$ be time-dependent and transform the background gauge field as $A^i \to A^i + \partial_t \epsilon^i$. Then, we consider the Lagrangian

$$\mathcal{L}(A^i) = \frac{m}{2}\left(\left(\partial_t x^1 - A_t^1\right)^2 + \left(\partial_t x^2 - A_t^2\right)^2\right) + B x^1 \left(\partial_t x^2 - A_t^2\right) + B x^2 A_t^1. \tag{A.7}$$

The first three terms are natural and the last term is such that $\frac{\partial \mathcal{L}(0)}{\partial A^i} = -p_i$. Under the gauge transformation

$$\mathcal{L}(A^i) \to \mathcal{L}(A^i) + B\left(\epsilon^2 A_t^1 - \epsilon^1 A_t^2 + \epsilon^2 \partial_t \epsilon^1\right) + \partial_t \left(B x^2 \epsilon^1\right). \tag{A.8}$$

It is easy to see that there is no way to add local terms to (A.7) and make the Lagrangian or even the action gauge invariant. This is the sign of the 't Hooft anomaly.

A way to restore the gauge invariance of the background field is the following. We add a bulk direction $b$, extend the gauge fields $A^i$ to the bulk such that the field strengths in the bulk,

$$F_{bt}^i = \partial_b A_t^i - \partial_t A_b^i, \tag{A.9}$$

vanish. Instead of imposing this vanishing condition, we can add two classical bulk fields $C^i$, which transform under the gauge symmetry as $C^i \to C^i + \epsilon^i$ and consider the bulk Lagrangian

$$\mathcal{L}_{bulk}(A^i, C^i) = B\left(C^1 F_{bt}^2 - C^2 F_{bt}^1 + A_b^1 A_t^2 - A_t^1 A_b^2\right). \tag{A.10}$$

(Note that if we integrate over $C^i$, this is the same as imposing $F_{bt}^i = 0$.) Under the gauge transformation, it transforms as

$$\mathcal{L}_{bulk}(A_i, C^i) \to \mathcal{L}_{bulk}(A_i) + B\left(\partial_b\left(\epsilon^1 A_t^2 - \epsilon^2 A_t^1 - \partial_t \epsilon^1 \epsilon^2\right) + \partial_t\left(A_b^1 \epsilon^2 - A_b^2 \epsilon^1 + \partial_b \epsilon^1 \epsilon^2\right)\right). \tag{A.11}$$

Ignoring the total time derivatives in (A.8) and (A.11), we see that the surface term in (A.11) cancels the gauge variation in (A.8).[4]

## A.2 A charged particle in $\mathbb{T}^2$ with a constant magnetic field

### A.2.1 A classical particle

Next, we consider the particle on $\mathbb{T}^2$. We use the discussion for $\mathbb{R}^2$ (Appendix A.1) and identify the coordinates

$$\begin{aligned} x^1 &\sim x^1 + \ell^1, \\ x^2 &\sim x^2 + \ell^2. \end{aligned} \tag{A.12}$$

This can be interpret as gauging

$$G_{gauge} = \mathbb{Z} \times \mathbb{Z} \subset G = \mathbb{R} \times \mathbb{R}. \tag{A.13}$$

The Lagrangian (A.1) is not single-valued under the identification (A.12). We can phrase it as not being $G_{gauge}$ gauge invariant. However, since it transforms by a total derivative, the classical equations of motion (A.2) are gauge invariant. Therefore, there is no problem in gauging $G_{gauge} = \mathbb{Z} \times \mathbb{Z}$.

The effect of the gauging is to reduce the global symmetry[5]

$$G = \mathbb{R} \times \mathbb{R} \quad \to \quad G_{classical} = U(1) \times U(1). \tag{A.14}$$

---

[4]Note that the anomaly theory (A.10) does not satisfy the common definition of an invertible anomaly theory, which is up to continuous deformations, nor is it a pure gauge theory.

[5]More generally, when we gauge $G_{gauge} \subset G$, the global symmetry is reduced to $G_{classical} = \frac{N_G(G_{gauge})}{G_{gauge}}$ with $N_G(G_{gauge})$ the normalizer of $G_{gauge}$ in $G$. In higher dimensions, gauging can also lead to new symmetries. In fact, even in quantum mechanics, when $G_{gauge}$ is discrete, as in our case, gauging can lead to a "$-1$-form symmetry" whose gauge field is a $\theta$-parameter [22, 27, 28].

### A.2.2 A quantum particle

In the quantum theory, the fact that the Lagrangian (A.1) is not single-valued under the identification (A.12) is more significant than in the classical theory and it leads to known consequences.

There are several ways to study the quantization of this Lagrangian. First, using canonical quantization we have to make sure that the operators acting on our Hilbert space are well-defined. Alternatively, in a Euclidean path integral presentation, one can define the exponential of the Euclidean action using differential cohomology.[6] Or, as in the spirit of this note, we can simply use (2) with $A^1 = 2\pi \frac{x^1}{\ell^1}$ and $A^2 = 2\pi \frac{x^2}{\ell^2}$.

The result of this more careful analysis shows that the magnetic field has to be quantized

$$B = \hbar F_{12} = 2\pi \frac{\hbar k}{\ell^1 \ell^2}, \qquad k \in \mathbb{Z}. \tag{A.15}$$

Also, the system depends not only on the magnetic field, but also on the holonomies of the background gauge field around the cycles of the torus

$$\exp\left(i \oint dx^1 A_1\right) = \exp\left(2\pi i k \frac{x^2 - x_0^2}{\ell^2}\right),$$

$$\exp\left(i \oint dx^2 A_2\right) = \exp\left(-2\pi i k \frac{x^1 - x_0^1}{\ell^1}\right). \tag{A.16}$$

(Compare with (28).) Here, $x_0^i$ are new parameters of the quantum theory. The existence of the two parameters $x_0^i$ is related to the fact that the classical symmetry $G_{classical} = U(1) \times U(1)$ is broken in the quantum theory to

$$G_{quantum} = \mathbb{Z}_k \times \mathbb{Z}_k. \tag{A.17}$$

We will now derive these well-known conclusions from a symmetry/anomaly perspective.

The quantum theory on $\mathbb{R}^2$ has a global $G = \mathbb{R} \times \mathbb{R}$ symmetry and we would like to gauge a subgroup $G_{gauge} = \mathbb{Z} \times \mathbb{Z}$ of it. This subgroup is generated by

$$U^1(\ell^1) = \exp\left(\frac{i\ell^1 p_1}{\hbar}\right),$$

$$U^2(\ell^2) = \exp\left(\frac{i\ell^2 p_2}{\hbar}\right). \tag{A.18}$$

The 't Hooft anomaly in $G$ (A.6) means that the generators $U^1$ and $U^2$ commute only when $B$ is quantized as in (A.15). We interpret it as the statement that gauging $G_{gauge} \subset G$ is consistent only when $G_{gauge}$ is anomaly free.

Classically, the remaining global symmetry is $G_{classical} = U(1) \times U(1)$ (A.14). However, the 't Hooft anomaly (A.6) of $G$ means that $U^i(\epsilon^i) = \exp\left(\frac{i\epsilon^i p_i}{\hbar}\right)$ commute with $G_{gauge}$ only when $\epsilon^i \in \frac{\ell^i}{k}\mathbb{Z}$. Hence, the symmetry $G_{classical}$ is reduced and the symmetry of the quantum theory is generated by

$$T^i = U^i\left(\frac{\ell^i}{k}\right) = \exp\left(\frac{i\ell^i}{\hbar k}p_i\right). \tag{A.19}$$

---

[6]In more physical terms, we can choose a local trivialization, i.e., let $x^1$ and $x^2$ be real valued and let them "jump" at some Euclidean point $\tau_*$, such that $x^i(\tau_*^+) = x^i(\tau_*^-) + W^i \ell^i$ with $W^i \in \mathbb{Z}$, the winding numbers around the Euclidean time direction $\tau$. Then, in order to ensure independence of $\tau_*$, we add to the Euclidean action a "correction term" $iBW^1\ell^1 x^2(\tau_*)$. See [1] for a review and a list of references from different points of view.

Using the identification (A.12), the symmetry group in the quantum theory is reduced to (A.17). Equivalently, the conserved classical charges $p_i$ in (A.4) are not single valued – they are not $G_{gauge} = \mathbb{Z} \times \mathbb{Z}$ gauge invariant. But exponentiating them as in $T^i$ in (A.19) leads to gauge invariant operators.

We interpret the explicit breaking

$$G_{classical} = U(1) \times U(1) \quad \rightarrow \quad G_{quantum} = \mathbb{Z}_k \times \mathbb{Z}_k, \tag{A.20}$$

as due to an ABJ anomaly. It arises from the gauging of $G_{gauge} \subset G = \mathbb{R} \times \mathbb{R}$ and the 't Hooft anomaly in $G = \mathbb{R} \times \mathbb{R}$.

As in the ABJ chiral anomaly, one can discuss two different charges. $p^i$ is conserved, but not gauge invariant and $\tilde{p}^i = m\partial_t x^i$ is gauge invariant, but not conserved. Also, as in the ABJ chiral anomaly, this explicit symmetry breaking in the quantum theory is associated with two $\theta$-terms $\hbar\left(\theta^1 \frac{\partial_t x^1}{\ell^1} + \theta^2 \frac{\partial_t x^2}{\ell^2}\right)$, which are related to the choice of $x_0^i$ in (A.16). (See the comment in footnote 5.)

We note that we can start with the system on $\mathbb{T}^2$ and not view it as gauging $\mathbb{Z} \times \mathbb{Z}$ in $\mathbb{R}^2$. In that case, the reduction of the classical global symmetry (A.20) can still be viewed as an ABJ anomaly. Here, we use the definition of the ABJ anomaly as the explicit breaking of a classical symmetry due to quantum effects.

Finally, the 't Hooft anomaly in $G$ (A.6), also leads to an 't Hooft anomaly in $G_{quantum}$

$$T^1 T^2 = \exp\left(\frac{2\pi i}{k}\right) T^2 T^1. \tag{A.21}$$

This quantum translation symmetry is used often in the body of this note.

## A.3 Summary and lessons

Let us recall the standard definitions of anomalies.

- An 't Hooft anomaly in a global symmetry $G$ is the quantum obstruction to gauging it. In quantum mechanics (as opposed to higher-dimensional quantum field theory), it is the statement that the symmetry group is realized projectively.

- An ABJ anomaly in a symmetry of a classical system $G_{classical}$ is its explicit breaking in the quantum theory to a subgroup $G_{quantum}$. This breaking arises from a careful definition of the quantum theory. Typically, this happens when we start with a global symmetry $G$ with 't Hooft anomaly and we gauge an anomaly free subgroup of it $G_{gauge} \subset G$. Classically, the global symmetry is $G_{classical}$, but due to the 't Hooft anomaly in $G$, the symmetry of the quantum theory after gauging is reduced to a subgroup $G_{quantum} \subset G_{classical}$.

The simple example above demonstrates these aspects of anomalies and highlights the following conclusions.

- The existence of 't Hooft anomaly does not signal an inconsistency of the theory. It is a property of the symmetry in the quantum theory. It should be clarified that it is inconsistent to gauge a symmetry $G$ with an 't Hooft anomaly. However, there is no problem gauging an anomaly free subgroup $G_{gauge} \subset G$.

- There is no need to add a "bulk" to make sense of a theory with an 't Hooft anomaly. Sometimes such a bulk is a convenient way to describe the anomaly. And in some cases, it is even physical. But it is not essential.

- In some cases, anomalies are related to quantum divergences and the need to regularize them, or to subtleties in the measure of the functional integral. Often, such subtleties are associated with the presence of fermions. But as the example above demonstrates, these sources of anomalies are not the only ones.

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
