# Peer review of "Anomalous Continuous Translations"

_SciPost Physics, doi:SciPost Phys. 19, 031 (2025)_

## Round 2 · Referee Report · Anonymous (Referee 1) · 2025-7-13

Strengths

The paper presents a very streamlined discussion on the breaking of continuous translation symmetries in QFT to discrete (non-abelian) translation symmetries due to the presence of nontrivial background gauge field and an ABJ-like anomaly. Along the way, the author addresses puzzles concerning such discrete translation symmetries and their anomalies which were studied previously.

The author's approach gives a unified perspective on this phenomena using a Chern-Simons type action for several different classes of models (e.g. abelian gauge theories and non-linear sigma models) and also connects to modern perspectives on symmetries and anomalies.

The presentation is very pedagogical: the author is able to convey the message without getting into the details of differential cohomology (which is the proper math framework for questions here but can be overwhelming for many physicists).

Weaknesses

None

Report

The journal's acceptance criteria are met and I recommend publication in SciPost after addressing the small changes below.

Requested changes

Here I list some small changes to be made (mostly typos):

  1. (3.10) is repetitive with (3.7). The author may just want to refer to the d=1 case of (3.7) here and comment on the general case in words (already given in (3.7)).
  2. end of second paragraph on pg 15: "of-shell"
  3. first paragraph in conclusion: "one turns one"
  4. obvious typo in (A.4) second equation
  5. (A.8): "=->"
  6. p25 second bullet point: "Gcalssical"

Recommendation

Publish (surpasses expectations and criteria for this Journal; among top 10%)

  • validity: top
  • significance: high
  • originality: high
  • clarity: top
  • formatting: excellent
  • grammar: excellent

Author:  Nathan Seiberg  on 2025-07-13  [id 5640]

(in reply to Report 1 on 2025-07-13)

I thank the referee for the positive comments and for catching all these typos. All of them will be fixed.

---

## Round 2 · Referee Report · Anonymous (Referee 2) · 2025-7-14

Strengths

Original, pedagogical, timely

Weaknesses

None.

Report

This paper presents a short but very pedagogical and insightful discussion of anomalies of continuous translation symmetries - how these can be in some cases broken to a discrete symmetry in the quantum theory. The author avoids use of fancy mathematical language, and provides enjoyable examples in appendices. Moreover relationships with other aspects - in particular the LSM theorem - are discussed and inspiring ideas put forward. Overall, this is a very nice, very smart paper. I noticed a few misprints, already mentioned by the first referee.

Recommendation

Publish (surpasses expectations and criteria for this Journal; among top 10%)

---

## Round 3 · Author Response

I thank the two referees for their comments. I have fixed the typos that were pointed out to me.

---

## Round 3 · List of Changes

I fixed the typos pointed out by the referee.

---

## Editorial Decision

published